# Blood Markers Show Neural Consequences of LongCOVID-19

**DOI:** 10.3390/cells13060478

**Published:** 2024-03-08

**Authors:** Norina Tang, Tatsuo Kido, Jian Shi, Erin McCafferty, Judith M. Ford, Kaitlyn Dal Bon, Lynn Pulliam

**Affiliations:** 1Department of Laboratory Medicine, San Francisco VA Health Care System, San Francisco, CA 94121, USA; norina.tang@va.gov (N.T.); tatsuo.kido@ucsf.edu (T.K.); erin.m.mccafferty@gmail.com (E.M.); 2Department of Neurology, San Francisco VA Health Care System, San Francisco, CA 94121, USA; jian.shi@ucsf.edu; 3Department of Neurology, University of California San Francisco, San Francisco, CA 94143, USA; 4Department of Mental Health, San Francisco VA Health Care System, San Francisco, CA 94121, USA; judith.ford@ucsf.edu (J.M.F.); kaitlyn.dalbon@ucsf.edu (K.D.B.); 5Department of Psychiatry and Behavioral Sciences, University of California San Francisco, San Francisco, CA 94143, USA; 6Department of Laboratory Medicine, University of California San Francisco, San Francisco, CA 94143, USA

**Keywords:** LongCOVID-19, blood markers, neuronal extracellular vesicles, BDNF, cortisol, cognition

## Abstract

Severe acute respiratory syndrome coronavirus 2 (SARS-CoV-2) persists throughout the world with over 65 million registered cases of survivors with post-COVID-19 sequelae, also known as LongCOVID-19 (LongC). LongC survivors exhibit various symptoms that span multiple organ systems, including the nervous system. To search for neurological markers of LongC, we investigated the soluble biomolecules present in the plasma and the proteins associated with plasma neuronal-enriched extracellular vesicles (nEVs) in 33 LongC patients with neurological impairment (nLongC), 12 COVID-19 survivors without any LongC symptoms (Cov), and 28 pre-COVID-19 healthy controls (HC). COVID-19 positive participants were infected between 2020 and 2022, not hospitalized, and were vaccinated or unvaccinated before infection. IL-1β was significantly increased in both nLongC and Cov and IL-8 was elevated in only nLongC. Both brain-derived neurotrophic factor and cortisol were significantly elevated in nLongC and Cov compared to HC. nEVs from people with nLongC had significantly elevated protein markers of neuronal dysfunction, including amyloid beta 42, pTau181 and TDP-43. This study shows chronic peripheral inflammation with increased stress after COVID-19 infection. Additionally, differentially expressed nEV neurodegenerative proteins were identified in people recovering from COVID-19 regardless of persistent symptoms.

## 1. Introduction

Due to declining new cases of severe acute respiratory syndrome coronavirus 2 (SARS-CoV-2) infections, the World Health Organization declared, on 5 May 2023, an end to the public health emergency of the COVID-19 pandemic. Unfortunately, the virus continues to persist with over 65 million registered cases of survivors with post-COVID-19 sequelae, also known as LongCOVID-19 (LongC) [1]. LongC can manifest in individuals of all different ages and backgrounds and is associated with the severity of the acute phase disease [2]. The estimated incidence rate is 50–70% for hospitalized patients and 10–30% for non-hospitalized cases [1]. The highest percentage diagnosis of LongC is in individuals aged 36–50 years old who are non-hospitalized with mild acute illness, as they represent the majority of COVID-19 cases [3]. Thirty-one percent of LongC cases have no identified preexisting chronic comorbidities, and females are more likely than males to be diagnosed with LongC [3].

LongC is thought to be a multi-organ syndrome with a broad array of clinical manifestations suggestive of underlying pulmonary, cardiovascular, endocrine, hematologic, renal, gastrointestinal, immunological and/or neurological disease [4]. LongC neurological sequelae (nLongC) can involve both the peripheral nervous system (PNS) and the central nervous system (CNS). PNS symptoms include muscle weakness, myalgias, hyposmia, hypogeusia, hearing loss/tinnitus, and sensorimotor deficits [5]. CNS symptoms include fatigue, brain fog, headache, sleep disorders, cognitive impairment, emotional/mood disorders, dizziness and dysautonomia [5]. Putative mechanisms for nLongC include neuroinflammation, autoantibody generation, microclot formation, blood vessel damage, and neuronal injury [1]. Currently, there are no established effective diagnostic tools or treatments for nLongC.

Extracellular vesicles (EVs) are nanoparticles secreted by most cells carrying molecular cargo that not only reflect the current physiological state of the secreting cell but also, upon uptake, can elicit responses in recipient cells that promote health or disease [6]. EVs can cross the blood–brain barrier (BBB) into the periphery, where they can serve as biomarkers of brain disease or as propagators of pathologic conditions. Alternatively, harmful EVs secreted by cells in the periphery can cross the BBB into the brain to facilitate neurological damage. EVs have been implicated in SARS-CoV-2 infection, LongC and nLongC [7,8]. Our lab investigations utilized neuronal-enriched EVs (nEVs), immunopurified from plasma with anti-L1CAM monoclonal antibodies, as vehicles and biomarkers of neurological diseases [9]. We previously reported on the plasma and nEV contents from people with nLongC at a time point 2–3 months post-infection. Compared to the plasma and nEVs obtained from pre-pandemic controls, we saw protein changes indicative of a peripheral cytokine reaction with differentially expressed neurodegenerative proteins [10].

In this study, we examined the plasma and nEVs of a nLongC cohort at a time point of around one year post-infection and compared a larger set of protein profiles to those individuals who have fully recovered from SARS-CoV-2 infection (Cov) as well as pre-COVID-19 controls. We found that both Cov and nLongC groups still had elevated IL-1β, while nLongC now showed elevated IL-8 in the plasma. In addition, plasma cortisol and brain-derived neurotrophic factor (BDNF) were elevated in both Cov and nLongC groups. A high proportion of the nEV proteins associated with neurodegeneration (Aβ42, FGF-21, KLK-6, pTau181, TDP-43, HMGB1) were significantly upregulated in nLongC, while only one was significantly upregulated in Cov (Aβ40). These results suggest a shift in the cytokine profile with a distinct set of injurious proteins present in some individuals that may contribute to altered nEV cargo and ongoing COVID-19 neurological sequelae.

## 2. Materials and Methods

### 2.1. Study Participants and Blood Collection

Volunteers with a documented history of SARS-CoV-2, as evidenced by a positive viral RNA PCR or antigen test result from nasal or throat swab, were recruited from the Veterans Affairs Health Care System, San Francisco and from advertising in social media from the San Francisco Bay Area. All volunteers were greater than 5 months post SARS-CoV-2 infection. All participants signed a written informed consent approved by the University of California, San Francisco Institutional Review Board. The nLongC subjects all had neurological complaints which included at least one of the following: difficulty with memory or concentration, or increased anxiety or depression. Exclusion criteria included past and present seizures, head trauma, loss of consciousness greater than 15 min, alcohol and/or substance abuse/dependence within 3 months of participation, HIV, and pregnancy. Cov participants were also greater than 5 months post infection with no complaints of residual infection. Fasting whole blood was collected in EDTA tubes between February 2022 and May 2023. Plasma and peripheral blood mononuclear cells (PBMCs) were collected and frozen in aliquots at −80 °C.

For the healthy controls (HC), frozen plasma collected from pre-pandemic individuals was purchased from Blood Centers of the Pacific (Vitalant, San Francisco, CA, USA) and stored at −80 °C until use.

### 2.2. APOE Genotyping

Frozen PBMCs were thawed in a 37 °C water bath and cells collected by centrifugation at 500× *g* for 10 min. Two million PBMCs were lysed with RLT Lysis Buffer (Qiagen, Germantown, MD, USA, catalog #1053393) and DNA isolated using the AllPrep DNA/RNA Mini kit (Qiagen, Germantown, MD, USA, catalog #80204). The PBMC DNA was added to TaqMan™ Fast Advanced Master Mix reagents (Thermo-Fisher Scientific, Inc., Waltham, MA, USA, catalog #4444554), amplified and subjected to TaqMan™ SNP Genotyping Assays for rs429358 and rs7412 (Thermo-Fisher Scientific, Inc., Waltham, MA, USA, PN 4351379) using an ABI ViiA 7 (Applied Biosystems Co., Waltham, MA, USA) instrument for endpoint PCR. Data were analyzed using TaqMan™ SDS software, version 1.2.1 (Thermo-Fisher Scientific, Inc., Waltham, MA, USA). A positive control for ε*3*/ε*4* (Coriell Institute for Medical Research, Camden, NJ, USA, catalog #NG11755) was included in the assay.

The *APOE* gene is polymorphic at the two nucleotides rs429358 and rs7412, resulting in 6 possible genotypes: ε*2/*ε*2*, ε*2/*ε*3*, ε*2/*ε*4*, ε*3/*ε*3*, ε*3/*ε*4* and ε*4/*ε*4* [11].

### 2.3. Plasma Multiplex Cytokine Analysis

Seven plasma cytokines (IL-1β, IL-4, IL-6, IL-8, IL-10, TNFα and IFNγ) were measured using the V-PLEX Viral Panel 2 Human Kit multiplex assay from Meso Scale Discovery (MSD) (Rockville, MD, USA, catalog #K15346D-1) per manufacturer’s instructions. Detection ranges for the different analytes were as follows: IL-1β, 0.149–610 pg/mL; IL-4, 0.0515–211 pg/mL; IL-6, 0.168–690 pg/mL; IL-8, 0.149–612 pg/mL; IL-10, 0.0869–356 pg/mL; TNFα, 0.0896–367 pg/mL and IFNγ, 0.366–1500 pg/mL. All assays were performed in duplicate, and the results read using a QuickPlex SQ 120 instrument (MSD). Analyses were performed using DISCOVERY WORKBENCH^®^ 4.0 software (MSD).

### 2.4. Plasma C-Reactive Protein (CRP), Cortisol, and BDNF Analyses

Plasma CRP (high sensitivity, CRPH) and cortisol were run at the San Francisco VA Health Care Clinical Laboratory. Cortisol was run using the UniCel DxI 600 Access Immunoassay System from Beckman Coulter (Brea, CA, USA) and CRP was run using the AU5800 Clinical Chemistry Analyzer, also from Beckman Coulter (Brea, CA, USA). 

Since platelets can release stored brain-derived neurotrophic factor (BDNF) upon activation, we used platelet poor plasma (PPP) in the BDNF assays. PPP was prepared by centrifuging the EDTA plasma at 10,000× *g* for 10 min at 4 °C and collecting the clarified plasma. Both pro- and mature BDNF in the PPP were measured using the Total BDNF Quantikine ELISA kit available from bio-techne R&D Systems (Minneapolis, MN, USA, catalog # BDNT00) per manufacturer’s instructions.

### 2.5. nEV Isolation

Neuronal-enriched extracellular vesicles (nEVs) were purified from frozen plasma as previously described [10] with modifications. Briefly, 2.5 units of thrombin (Sigma-Aldrich, Burlington, MA, USA, catalog #T4648) was added for 1 h to 500 μL of plasma to remove fibrils and coagulation proteins, followed by centrifugation at 6000× *g* for 20 min. Total EVs (tEV) were then polymer precipitated by adding 252 μL of ExoQuick^TM^ Exosome Precipitation Solution (Systems Biosciences, Palo Alto, CA, USA; catalog #EXOQ20A-1) to the clarified plasma in the presence of 3× protease and phosphatase inhibitors (Thermo-Fisher Scientific, Rockford, IL, USA, catalog #78446). The precipitated tEV pellets were resuspended in 700 μL water containing 1× protease and phosphatase inhibitors. Four micrograms of biotinylated L1CAM mouse monoclonal antibody (Clone 5G3, Thermo-Fisher Scientific, Rockford, IL, USA, catalog #13-1719-82) in 3% BSA was added for 2 h at room temperature followed by addition of 80 μL of streptavidin-conjugated agarose beads (Pierce^TM^ Streptavidin Plus UltraLink^TM^ Resin from Thermo-Fisher Scientific, Rockford, IL, USA, catalog #53117) in 3% BSA for 1 h at RT. The L1CAM+ EV agarose bead complexes were washed and 200 μL of 0.1 M Glycine-HCl (pH3) added for 10 s on ice followed by centrifugation at 4500× *g* for 5 min at 4 °C. Supernatants containing the released nEVs were collected, neutralized with 30 μL 1 M Tris-HCl (pH8), aliquoted and stored at −80 °C until analysis.

### 2.6. Nanoparticle Tracking Analysis (NTA)

NTA was performed on intact nEV samples to determine size and particle counts. Data were generated using a ZetaView^®^ ×30 QUATT instrument (Particle Metrix GmbH, Inning am Ammersee, Germany) with a 488 nm laser-equipped chamber. Instrument calibration was performed using a known concentration of 100 nm polystyrene beads. Eleven positions were counted and analyzed using ZetaView version 8.05.16 SP3 software. Each position was measured for 2 cycles using a sensitivity of 75 and a shutter value of 100. Triplicate readings were performed for each sample and the average reported.

### 2.7. Tetraspanins Multiplex Assay

Intact nEVs were quantified for tetraspanin protein levels using MSD assays for CD9, CD63 and CD81 (MSD, Rockville, MD, USA, custom kit catalog #K15228N-1 using antibody set catalog #s F215M-3-8, F2215L-3/-8 and F215N-3-8). All MSD assays were performed according to manufacturer’s instructions and each sample was tested in duplicate. All analyses were performed using a QuickPlex SQ 120 instrument (MSD) and DISCOVERY WORKBENCH 4.0^®^ software.

### 2.8. EV Lysate Preparation

Immediately after isolation (see Section 2.5 above), a portion of the nEVs were lysed for protein analyses using 2× volume of lysis buffer containing a final concentration of 0.25% BSA, 2× protease and phosphatase inhibitors, and M-PER^TM^ Mammalian Protein Extraction Reagent (Thermo-Fisher Scientific, catalog #78501). Lysates were freeze-thawed twice, aliquoted and stored at −80 °C until analyses.

### 2.9. Enzyme-Linked Immunosorbent Assay (ELISA) on nEV Lysates

ELISAs were performed on the nEV lysates to determine protein levels for Alix (ALG-2-interacting protein X, also known as programmed cell death 6-interacting protein), high mobility group box 1 (HMGB1) and NeuN (neuronal nuclei protein, also known as RNA binding protein fox-1 homolog 3 [RBFOX3]) using the following commercially-available kits: Alix (Lifeome, Oceanside, CA, USA, catalog #CSB-EL017673HU, range: 0.031–2 ng/mL), HMGB1 (Novus Biologicals, LLC, Centennial, CO, USA, catalog #NBP2-62766, range: 0.031–2 ng/mL) and NeuN (Abbexa LLC, Houston, TX, USA, catalog #abx541982, range: 0.312–20 ng/mL). All ELISAs were performed according to manufacturer’s instructions and each sample was analyzed in duplicate. Protein concentrations were determined by absorbance using a SpectraMax M5 plated reader (Molecular Devices, LLC, San Jose, CA, USA) with Softmax Pro 7 software (Molecular Devices).

### 2.10. Luminex Multiplex Assay for Neurodegeneration Proteins

Luminex bead assays were performed on the nEV lysates in duplicate using a Neurodegeneration 9-plex Human ProcartaPlex^TM^ panel available from ThermoFisher (catalog #EPX090-15836-901) following manufacturer’s directions. The 9 neurodegeneration protein targets in the multiplex assay were amyloid beta (Aβ) 1–40 (Aβ40, detection range: 451–1,847,000 pg/mL), Aβ 1–42 (Aβ42, range: 0.42–1700 pg/mL), fibroblast growth factor 21 (FGF-21, range: 8.64–35,400 pg/mL), kallikrein-related peptidase 6 (KLK-6, range: 5.57–22,800 pg/mL), neural cell adhesion molecule 1 (NCAM-1, range: 54–221,900 pg/mL), neurogranin (NRGN, range: 10–41,600 pg/mL), p-T181-tau (pTau181, range: 1.95–2000 pg/mL), TAR DNA binding protein 43 (TDP-43, range: 96–393,200 pg/mL), and total Tau (tTau, range: 20–80,000 pg/mL). Results were read using a Luminex LX200 instrument with Luminex xMAP Technology. Data were analyzed using Milliplex^TM^ Analyst version 5.1 software.

### 2.11. Statistical Analysis and Bioinformatics

All statistical analyses were carried out using GraphPad Prism 10 software (GraphPad Software Inc., Boston, MA, USA). Since the data values were not distributed normally, the Kruskal Wallis test followed by post-hoc Dunn’s test was used when comparing the group medians between three groups (Figures 2A, 3A, 4A–C and 5 all graphs except HMGB1) and the Mann–Whitney U test was used when comparing the group medians between two groups (Figure 5 HMGB1 graph). For correlation analyses, the Spearman 2-tailed correlation was used.

Several computational tools and software were used to classify and detect blood proteins associated with nLongC. Data analyses were carried out using MATLAB (Version R2023a, The MathWorks, Inc., Natick, MA, USA) and R (Version 4.2.2; R Core Team). Gene Ontology (GO) enrichment analysis was performed using the online platform provided by the Gene Ontology Consortium (http://geneontology.org/ (accessed on 9 February 2024)) to explore the biological implications of the identified genes. For functional annotation, Database for Annotation, Visualization, and Integrated Discovery (DAVID) (https://david.ncifcrf.gov/home.jsp/ (accessed on 9 February 2024)) and g:Profiler (ELIXIR, Tartu, Estonia; https://biit.cs.ut.ee/gprofiler/gost/ (accessed on 9 February 2024)) were used.

## 3. Results

### 3.1. Participant Demographics and Clinical Data

This study involved 45 COVID-19 participants who tested positive by nasal or throat swab for SARS-CoV-2 virus using nucleic acid PCR test or rapid antigen test. These 45 participants were clinically assessed >5 months post-infection, with a range of 5–24 months, between February 2022 and May 2023. None of the participants were hospitalized. Infection over this time involved at least five different viral variants (Figure 1). Of the 45 COVID-19 participants, 12 did not report any lingering COVID-19 symptoms (Cov), while 33 complained of lingering or worsening COVID-19 neurological symptoms (nLongC) at the time of fasting blood draw and clinical assessment. For comparative analyses, age- and sex-matched plasma were obtained from the local blood bank from individuals who donated blood in the year 2019 prior to the COVID-19 pandemic (HC) (Table 1).

Seventy percent of the study participants were males (51 out of 73) while thirty percent were females (22 out of 73) (Table 1). There were no significant differences between the groups in terms of sex, race, ethnicity, *APOE* genotype, comorbidities, and vaccination status. The majority of COVID-19 participants were infected after vaccination. The average age of the study participants was 47 years for HC (range 20–67 yrs), 39 years for Cov (range 24–65 yrs) and 45 years for nLongC (range 23–65 yrs). The average time between infection and blood draw was 297 days for Cov (range 147–765 days) and 311 days for nLongC (range 96–1048 days).

### 3.2. Plasma Cytokines

Levels of peripheral markers of inflammation were determined using a seven multiplex MSD cytokine array (Figure 2). There were no differences in IFN-γ, IL-4, IL-6, IL-10 and TNF-α between the 3 groups. There was an increase in IL-8 in nLongC (4.8 pg/mL) compared to Cov (4.2 pg/mL) and HC (3.2 pg/mL) and a significant increase in IL-1β in both nLongC (0.08 pg/mL) and Cov (0.09 pg/mL) compared to HC (0.02 pg/mL) (Figure 2A). According to receiver-operating characteristic (ROC) analysis, IL-1β showed an area under the curve (AUC) of 0.82 (*p* < 0.0001) for nLongC and an AUC of 0.92 (*p* < 0.0001) for Cov (Figure 2B, left panel, red lines). In contrast, IL-8 was predictive for nLongC (AUC = 0.70, *p* = 0.01) but not for Cov (AUC = 0.56, *p* = 0.58) (Figure 2B, right panel, black lines).

To further investigate neurological symptoms in nLongC, we measured plasma BDNF, CRP and cortisol. To determine the level of systemic inflammation, we measured CRP and found no difference between the groups (Figure 3A), showing a discordance between peripheral IL-1β, which was elevated, and CRP. BDNF is a neurotrophin associated with synaptic plasticity and is important in memory and cognition [14]. Decreases in BDNF are associated with aging and may contribute to neurodegeneration and cognitive decline while increases may represent a transient compensatory response [15]. There have been inconsistent reports on BDNF in COVID-19 infection. Hospitalized patients with COVID-19 and those with neurological manifestations had lower serum BDNF than healthy controls [16,17] while those recovering from COVID-19 infection had higher BDNF levels [18]. In this study, both nLongC and Cov participants had significantly elevated BDNF compared to HCs (Figure 3A). High stress has been associated with poor cognitive function [19]. We showed significant elevation in cortisol levels in all COVID-19 subjects compared to healthy controls with no difference between those recovered from infection and those with neurological manifestations (Figure 3A). For nLongC, there was a positive correlation between BDNF and IL-1β (Figure 3B), which may be indicative of continued inflammation in these individuals. Unlike IL-1β, cortisol showed a negative correlation to BDNF (Figure 3C). This inverse relationship between BDNF and cortisol has been seen in people under stress [20]. For Cov, there was a negative correlation between cortisol and time after infection (Figure 3D), which may continue to decrease with time as these people continue their recovery.

### 3.3. nEV Characterization

We characterized nEVs from the three cohorts using several methods including nanoparticle tracking analysis (NTA) for size and concentration (Figure 4A). nEV sizes ranged from 136–178 nm for all three groups. Both HC and Cov groups had a mean nEV size of 158 nm while nLongC showed larger-sized nEVs with a mean of 164 nm. nEVs were most numerous in the nLongC group with a median of 21.1 × 10^8^ particles/mL plasma versus 11.5 × 10^8^ particles/mL for Cov and 6.6 × 10^8^ particles/mL for HC. Transmembrane tetraspanins on the nEVs were elevated in Cov and nLongC, compared to HC (Figure 4B), consistent with the increased particle counts observed in Figure 4A. For CD9, the levels were 4 × 10^−3^ AU/mL (HC), 259 × 10^−3^ AU/mL (Cov) and 572 × 10^−3^ AU/mL (nLongC). For CD63, the levels were 1.5 × 10^−4^ AU/mL (HC), 96.4 × 10^−4^ AU/mL (Cov) and 183.5 × 10^−4^ AU/mL(nLongC). For CD81, the levels were 1.5 × 10^−4^ AU/mL (HC), 14.0 × 10^−4^ AU/mL (Cov) and 24.3 × 10^−4^ AU/mL (nLongC). Alix, an exosome marker, was increased in both the Cov and nLongC groups (129 and 137 pg/mL, respectively versus 25 pg/mL for HC), again consistent with the increased particle counts (Figure 4C). Levels of NeuN protein, a neuron marker, were similar between the groups (Figure 4C). Since the nEV counts, tetraspanins and Alix were all increased in the Cov and nLongC groups, we normalized EV protein cargo to plasma volume in subsequent protein analyses.

### 3.4. nEV Protein Markers

EVs carry a diverse cargo reflective of the health of the parent cell. By selecting for neuronally-enriched EVs, there is an opportunity to determine how neuron dysfunction could manifest neurological symptoms. We lysed nEVs and analyzed the proteins using a 9-plex neurodegenerative assay (Luminex) and an ELISA for HMGB1 (Figure 5). Due to sample limitations, we were not able to test for HMGB1 in the Cov group. Protein analytes were normalized to plasma volume. All proteins tested, except Aβ40, were significantly elevated in nEVs from people with nLongC‚ compared to HC. In nEVs from the Cov group, Aβ40 was significantly elevated compared to HC. nEVs from people with nLongC had Aβ40 levels similar to HC. tTau, NRGN and NCAM-1 were significantly elevated above HCs in both the Cov and nLongC groups. nEVs from the nLongC group had significantly elevated neuronal proteins compared to nEVs from HC, including Aβ42, FGF-21, KLK-6, NCAM-1, NRGN, pTau181, TDP-43, tTau and HMGB1.

### 3.5. Blood Markers and Informatic Analysis

To determine biological markers that differentiate nLongC from Cov and HC samples, we screened and utilized multiple omics-derived markers including plasma and nEV cargo. Figure 6A summarizes the proteins in the Cov and nLongC groups that were found to be differentially expressed compared to the HC cohort. There were 10 differentially expressed biomolecules shared by both Cov and nLongC (IL-1β, CD9, CD63, CD81, Alix, cortisol, BDNF, NCAM-1, NRGN, tTau), 1 unique to Cov (Aβ40), and 6 unique to nLongC (IL-8, Aβ42, FGF-21, KLK-6, pTau181, TDP-43). The heatmap (Figure 6B) visualizes the differential expression levels of selected markers across the three cohorts, with red intensity indicating high expression levels. The intensity increases going from left to right with the exception of Aβ40 (bottom row), which was most intense in nEVs from the Cov group.

To elucidate the functional meanings of these markers, we employed bioinformatics tools DAVID and g:Profiler, which identified 15 significant Gene Ontology (GO) terms related to biological processes (BP) and Human Phenotype Ontology (HP) terms associated with Alzheimer’s disease (AD). These terms are listed in Figure 6C, where the chord diagram shows the association between the markers and GO and HP terms. Among these identified terms, GO:1990000 (amyloid fibril formation), HP:0000726 (Dementia), and HP:0002185 (neurofibrillary tangles), shown in red, are particularly notable for their direct links to AD. This reinforces our previous findings [9,10] and further confirms the link between these markers and AD pathology. Interestingly, it also suggests that nLongC may accelerate the development of AD or promote its progression. Additionally, GO terms such as GO:0033554 (cellular response to stress) and GO:0006950 (response to stress), shown in red text, highlight potential pathophysiological aspects of nLongC. Furthermore, the identification of GO:0002274 (myeloid leukocyte activation), shown in red text, suggests the activation and accumulation of myeloid cells in the brain, possibly contributing to prolonged neuroinflammation in nLongC patients. This is particularly relevant as brain myeloid cells, including macrophages and microglia, are known to play critical roles in the development and progression of AD [21]. The identification of GO:0048143 for astrocyte activation, highlighted in red text, implicates not only the important role astrocytes play in the CNS innate immune response to SARS-CoV-2 infection but also their potential role as a viral reservoir in nLongC [22].

## 4. Discussion

This was a pilot study with a small sample size of 28 pre-pandemic COVID-19 controls, 12 recovered individuals, and 33 subjects with self-reported persisting neurological symptoms with a focus on blood biomolecules that are differentially expressed in nLongC. Others have reported female sex and comorbidity presence as being risk factors for LongC [23,24]. In our cohort, these risk factors did not reach statistical significance (Table 1).

We previously analyzed the plasma and nEVs of nLongC subjects who were 36–103 days post-infection (average 77 days) with, likely, the 614 SARS-CoV-2 variant given the time of infection (early 2020) [10]. These early nLongC subjects showed elevations in IL-1β, IL-4 and IL-6. The elevated IL-1β finding here in all COVID-19-infected individuals, regardless of lingering sequelae (Figure 2A), is consistent with other published reports [18,25,26]. In contrast, our current cohort, who were infected later and who experienced nLongC for a longer time (mean of 311 days), showed a different plasma cytokine profile with elevations in IL-1β and IL-8. The shift from IL-4 and IL-6 to IL-8 may be a result of the two different cohorts being infected with different viral variants, vaccination response or timing to recovery. The increased plasma IL-1β and IL-8 found in this study have also been reported in a larger LongC cohort with a mean post-infection time of 140 days [27]. In the elderly, elevated IL-8, and not IL-1β, has been associated with poor memory, speed domains and motor functions [28]. In addition, higher plasma IL-8 levels were found in patients with AD that correlated with cognition severity [29]. Because others have found similar pathological elements between SARS-CoV-2 infection and AD [30,31], it is tempting to speculate that increased IL-8 may play an important role in the neurological sequelae of COVID-19.

Cytokines, as well as CRP, can be measures of acute inflammation and have been found to be elevated in early COVID-19 infection [32]. Published results comparing CRP levels between LongC and those people recovered from COVID-19 vary. There are many reasons for this including sampling criteria, limitation in protein discovery and hospitalization. One review, based on literature studies, suggested that overexpression of IL-6, CRP and TNFα for one or more months post-infection may constitute a core set of biomarkers for LongC even though when compared to recovered patients, only 6/10, 3/20 and 3/20 studies found increased IL-6, CRP and TNFα, respectively [33]. Our present study found no increase in peripheral IL-6, TNFα or CRP (Figure 2A and Figure 3A). This may be a reflection that LongC is not a continuation of an acute inflammatory condition.

Under normal conditions, cortisol binds to the glucocorticoid receptor (GR), eliciting anti-inflammatory effects. However, prolonged or excessive cortisol may result in a compensatory down-regulation of or resistance to GR-mediated signaling, leading to pro-inflammatory effects [34]. In the present study, cortisol was significantly elevated in both Cov and nLongC groups compared to HCs (Figure 3A), although the levels were still within the normal range of 10–20 μg/dL [35]. Over time after infection, we saw a decrease in cortisol in recovered COVID individuals, but not in people with nLongC (Figure 3D). Continued elevated cortisol levels in nLongC individuals may have negative cognitive effects. Changes in levels of cortisol have been reported in people at risk for mental illness, especially during the COVID-19 pandemic [35]. Other studies have shown an association of increased plasma cortisol with elevated Aβ burden as seen by PiB-PET imaging, which can discriminate between healthy control persons, those with mild cognitive impairment (MCI) and those with AD [36,37]. The fact that there was also an increase in Aβ42 in nEVs from people with nLongC that was not seen in nEVs from people who recovered from Cov (Figure 5) further supports the hypothesis that persistently elevated cortisol may facilitate the neurological sequelae seen in some COVID individuals by possibly increasing Aβ42 deposition. A recent publication by Klein et al. using a larger cohort of 275 individuals with or without LongC showed decreased cortisol in the LongC group compared to healthy, uninfected vaccinated controls and previously infected, vaccinated controls without persistent symptoms [27]. The difference in results between our study and the Klein study may be due to the different post-infection times of the LongC cohorts. In the Klein study, the average post-infection time was much longer at around 450 days, while in our study it was shorter at 311 days. Since injury to the adrenal glands has been reported in LongC individuals as a result of immune damage leading to hypocortisolemia [38], it is possible that the elevated cortisol levels we saw in our nLongC cohort will diminish over time as continued inflammation delivers more harm to the adrenal tissue.

Brain-derived neurotrophic factor is a neurotrophin that modulates synaptic plasticity and promotes neurogenesis [39,40]. It crosses the BBB and can be measured in the brain [41]. Since BDNF is a trophic factor, it is thought that a decrease in BDNF could contribute to neurodegeneration and, indeed, it is significantly decreased in people with AD [42]. However, further studies looking at people with MCI showed an increase in peripheral BDNF as a compensatory mechanism with early AD showing higher levels and late AD showing lower levels of BDNF when compared to controls [43,44]. In this study we found a moderate positive correlation of IL-1β to BDNF (Figure 3B) and a moderate negative correlation to cortisol (Figure 3C) in nLongC individuals. Since high cortisol reduces BDNF expression under stressful conditions [20], our data suggest that people with nLongC may be under chronic stress due to the systemic inflammation and may respond initially with a compensatory elevated level of BDNF that signals MCI. However, over time, BDNF may decline to a level that signals more severe neurological dysfunction and cognitive impairment. Since MCI is thought to be a continuum to AD, measuring BDNF further out in time in LongCOVID-19 may illuminate this pattern.

Neuronal enriched extracellular vesicles may be a good indicator of neuronal health in real time. They are less invasive than cerebrospinal fluid and less complex than total EVs from the periphery. Our lab is the first to study nEVs as biomarkers of LongC [10]. In our previous report on nLongC, we analyzed nEVs and found that the neurodegenerative proteins in a multiplex assay were all significantly increased in people with LongC, with or without neurological symptoms [10]. We did not test people without LongC. In the present study, it is clear there are differences between people who do not have lingering post-infection symptoms and those with nLongC. There are no differences in the levels of Aβ42, FGF-21, KLK-6, pTau181 and TDP-43 cargo in nEVs from the HC and Cov groups. However, people who had COVID-19 infection and recovered do have altered nEV cargo with increases in Aβ40, NCAM-1, NRGN, and tTau, suggesting that neurons are still affected. People recovered from COVID-19 infection also have increased IL-1β, cortisol and BDNF. These biomolecules are not normal and need to be followed longitudinally to see if they revert to normal. Most troubling are the sustained high levels of the other toxic proteins, Aβ42, FGF-21, KLK-6, pTau181 and TDP-43, observed in nEVs from nLongC individuals. When we reported this in nEVs of people with nLongC in 2021, we speculated that this secretion might be a positive mechanism for neurons to expel toxic proteins. This may still be a possibility, but at some point over time, this accumulation may have negative consequences on cognition.

HMGB1 is a nuclear protein that promotes inflammation [45] and we reported an increase in this protein previously with nEVs from nLongC individuals [10] and in HIV-infected individuals with cognitive impairment [46]. When released in the brain, it can activate microglia and when released in the periphery from EVs can promote inflammation [47].

The results in this study show a chronic peripheral inflammation in nLongC individuals as evidenced by elevated IL-1β and IL-8. In addition, there is increased stress after COVID-19 infection that persists in nLongC as indicated by the increased BDNF and cortisol levels. These pathological events may impact the nervous system and injure neurons as EVs released by neurons show differential expression of proteins that participate in neurodegeneration. 

There are several limitations to this study. Because the participant numbers are small, these results will need to be confirmed with a larger cohort. A rigorous assessment of cognitive dysfunction is missing as the participants’ mental status was self-reported. However, despite a lack of vigorous neurocognitive testing, there were still significant biological differences in the groups. We speculate that self-reporting of a change in cognition may precede any clinical neurological testing results. Going forward, a more standardized assessment of cognitive health should be used. People with pre-existing cognitive impairment or clinical psychiatric illness were not included. With these drawbacks, the data suggest emerging significant differences in several measurable parameters between healthy pre-pandemic controls and people recovering from COVID-19 and those with neurological complaints, suggesting a biological rationale for these conditions.

## 5. Conclusions

This study reports differentially expressed peripheral markers of inflammation and stress after SARS-CoV-2 infection between healthy pre-pandemic controls, people recovered from COVID-19 and neuroLongCOVID-19 subjects. Neurotoxic proteins associated with neurodegeneration were identified from neuronal enriched EVs that differed between the three groups. One or more of the proteins identified in this study have also been shown in other neurological disorders (e.g., AD, HIV-associated neurocognitive disorder, Parkinson’s disease, epilepsy, multiple sclerosis, Guillain–Barre syndrome, amyotrophic lateral sclerosis) with other etiologies (bacteria, fungus, other viruses) [9,48,49,50], suggesting common pathological mechanisms. It remains to be seen, perhaps with the use of machine learning algorithms, whether our specific panel of combined proteins uniquely identifies neuroLongCOVID-19 and, thus, may serve as biomarkers for this disease. In addition, longitudinal studies are needed to determine how these blood-based markers may be used to guide future studies into the immune and neurological mechanisms as well as diagnosis and treatment of nLongC.

## Figures and Tables

**Figure 1 cells-13-00478-f001:**
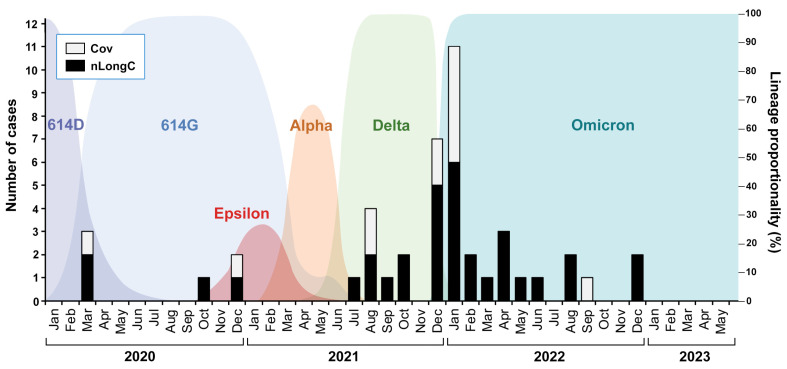
nLongC and Cov participants were infected over a period of three years with, likely, different SARS-CoV-2 variants. The COVID-19 cases described in this study are shown as black and white bars superimposed onto a background of SARS-CoV-2 variants described by Wang et al., 2022 [12] and Lam et al., 2020 [13].

**Figure 2 cells-13-00478-f002:**
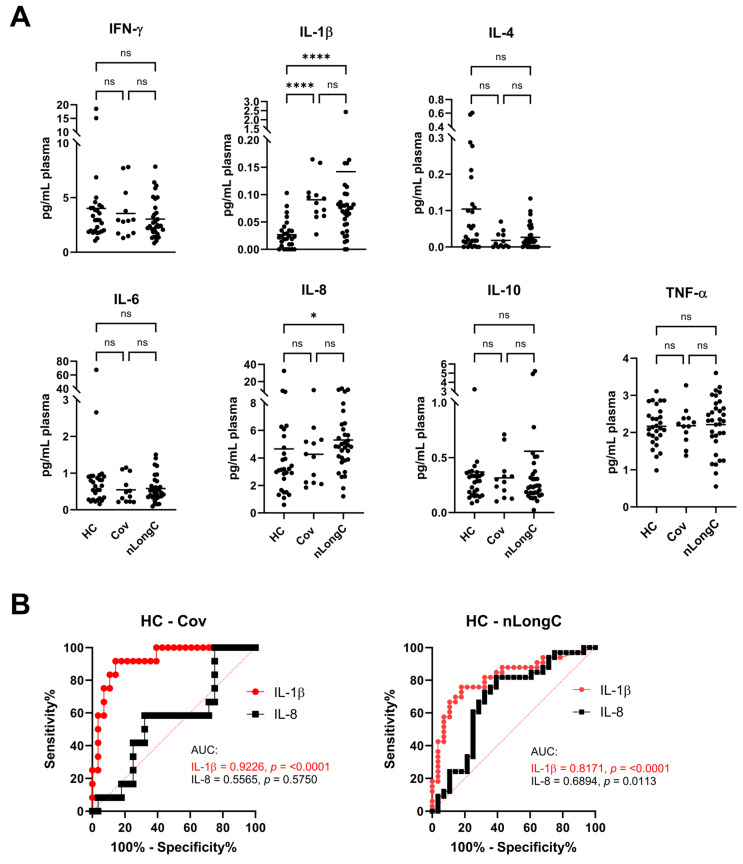
Plasma cytokines. (**A**) Meso Scale Discovery (MSD) analysis for seven cytokines showed a significant increase in IL-1β for both Cov and nLongC groups. IL-8 was increased only in the nLongC group. Horizontal bars indicate group medians which were compared using Kruskal–Wallis tests followed by post-hoc Dunn tests. *p* value symbols are as follows: ns *= p* > 0.05, * *p* ≤ 0.05) and **** *p* ≤ 0.0001. HC (*n* = 28), Cov (*n* = 12) and nLongC (*n* = 33). (**B**) Receiver-operating characteristic (ROC) curves for significantly differentially expressed plasma cytokines, Il-1β and IL-8. Area under curve (AUC) is shown in graphs. Significantly differentially expressed cytokines are displayed for each pair of groups: IL-1β (solid red line) and IL-8 (solid black line).

**Figure 3 cells-13-00478-f003:**
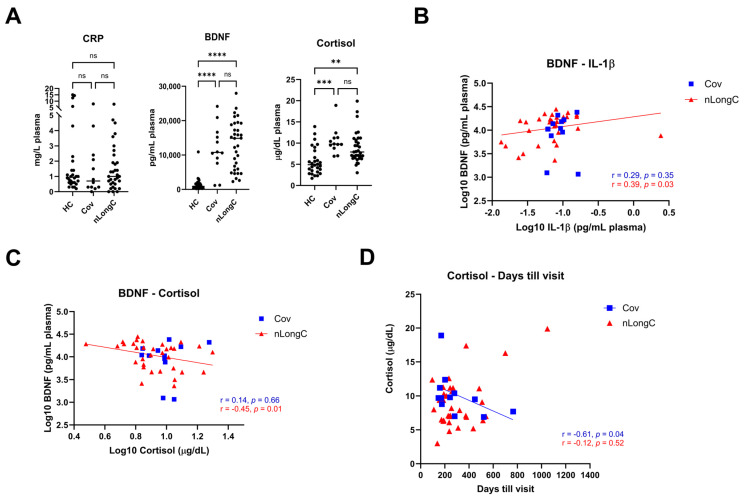
Plasma levels of brain-derived neurotrophic factor (BDNF), C-reactive protein (CRP) and cortisol. (**A**) People with Cov and nLongC had elevated levels of BDNF and cortisol without increased CRP (Kruskal–Wallis tests followed by post-hoc Dunn’s tests). ns = *p* > 0.05, ** *p* ≤ 0.01, *** *p* ≤ 0.001 and **** *p* ≤ 0.0001. BDNF had a moderate positive correlation with IL-1β in the nLongC group (**B**) (Spearman r = 0.39, *p* = 0.03) and a moderate negative correlation with cortisol (**C**) (Spearman r = −0.45, *p* = 0.0087). (**D**) Cortisol decreased over time in Cov subjects but not in nLongC individuals. For all panels, HC (*n* = 28), Cov (*n* = 12) and nLongC (*n* = 33).

**Figure 4 cells-13-00478-f004:**
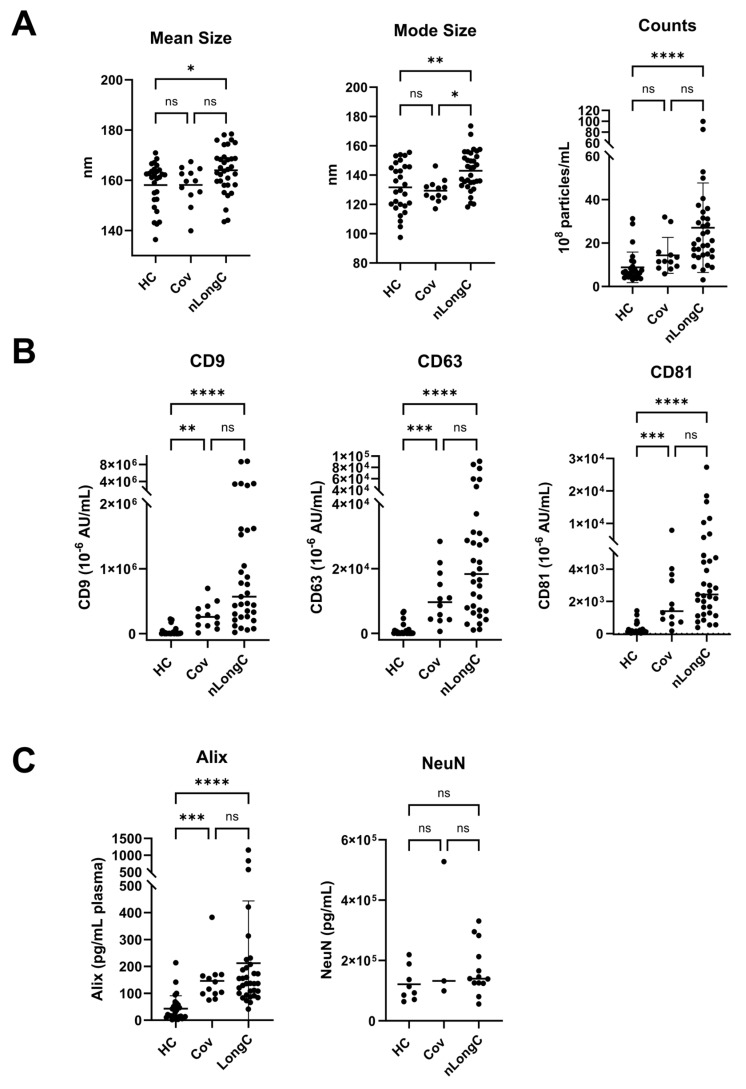
Characterization of nEVs. (**A**) Nanoparticle tracking analysis (NTA) showed larger and more abundant nEVs for nLongC. (**B**) nEV tetraspanins CD9, CD63 and CD81 were all increased in both Cov and nLongC groups. (**C**) Exosomal protein Alix was increased in both Cov and nLongC groups while NeuN was present in all nEVs at similar levels between the three groups. For NeuN, there were fewer values in each group due to insufficient sample availability: HC (*n* = 8), Cov (*n* = 3), nLongC (*n* = 13). Kruskal–Wallis tests followed by post-hoc Dunn’s tests were performed for all comparisons. ns = *p* > 0.05, * *p* ≤ 0.05, ** *p* ≤ 0.01, *** *p* ≤ 0.001 and **** *p* ≤ 0.0001. Group sample sizes for all (except NeuN) were HC (*n* = 28), Cov (*n* = 12) and nLongC (*n* = 33).

**Figure 5 cells-13-00478-f005:**
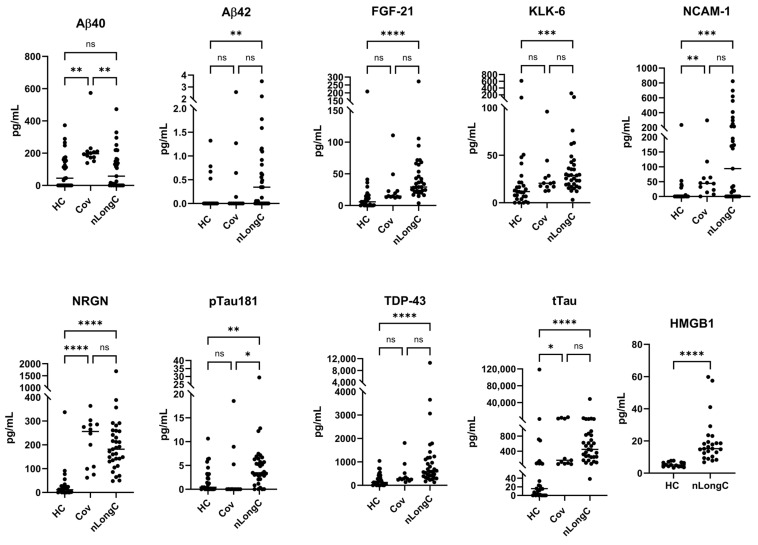
Neuronal protein cargo in nEVs. nEV lysates were analyzed using a 9-plex Luminex neurodegeneration panel and ELISA for HMGB1. Concentrations were normalized to plasma volume. All of the proteins except Aβ40 and tTau were increased in nEVs from the nLongC group. For the Cov group, only four proteins were elevated (Aβ40, NCAM-1, NRGN, tTau). Kruskal–Wallis tests followed by post-hoc Dunn’s tests were performed for all comparisons. ns = *p* > 0.05, * *p* ≤ 0.05, ** *p* ≤ 0.01, *** *p* ≤ 0.001 and **** *p* ≤ 0.0001. Group sample sizes were HC (*n* = 28), Cov (*n* = 12) and nLongC (*n* = 33), except for HMGB1 which were HC (*n* = 15) and nLongC (*n* = 25).

**Figure 6 cells-13-00478-f006:**
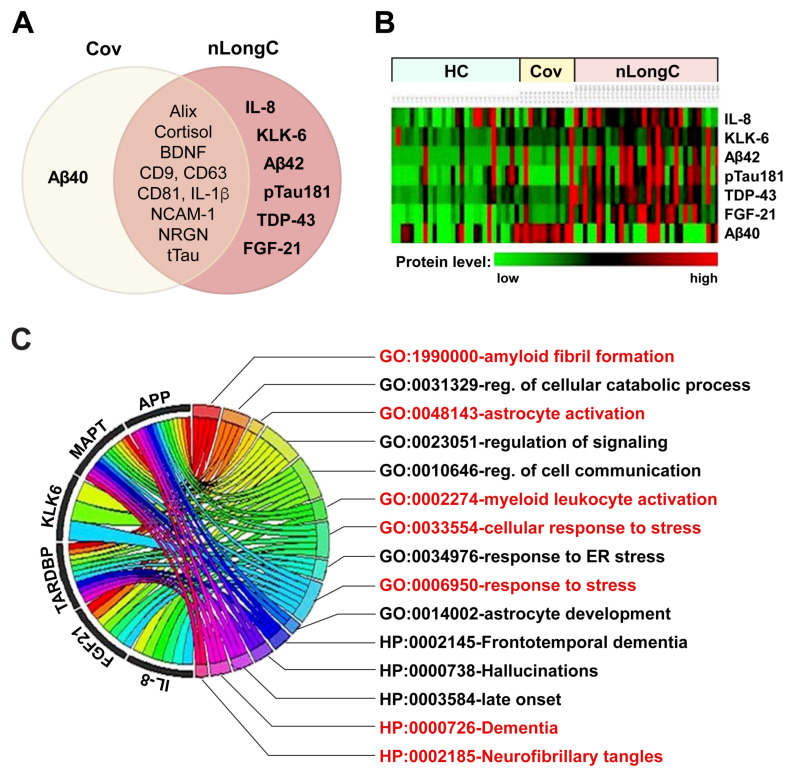
Blood biomolecules and informatics classification analyses for nLongC. (**A**) Venn diagram delineating the differentially expressed blood biomolecules identified in this study. (**B**) Heatmap of eight critical blood protein levels, with samples from healthy controls (HC) on the left, Cov in the middle, and nLongC on the right. Red represents higher expression and green represents lower expression. (**C**) Chord diagram depicting the connections between the genes and their associated GO and HP terms, providing insights into the biological processes and functions related to nLongC pathogenesis. GO and HP terms, along with their respective biological processes, indicated in red, represent potential pathological mechanisms of nLongC, some of which show associations to AD.

**Table 1 cells-13-00478-t001:** Demographics of healthy control, Cov and nLongC groups.

		HC (%) *n* = 28	Cov (%) *n* = 12	nLongC (%) *n* = 33	*p* Value
**Sex**	Female	5 (17.9)	3 (25.0)	14 (42.4)	0.1042
	Male	23 (82.1)	9 (75.0)	19 (57.6)	
**Race**	White	16 (57.1)	5 (41.7)	13 (39.4)	0.7034
	African American	7 (25.0)	3 (25.0)	8 (24.2)	
	Asian	1 (3.6)	2 (16.7)	5 (15.2)	
	More than one/Other	4 (14.3)	2 (16.7)	7 (21.2)	
**Ethnicity**	Hispanic	5 (17.9)	1 (8.3)	6 (18.2)	0.7089
	Non-Hispanic	23 (82.1)	11 (91.7)	27 (81.8)	
** *APOE* ** **genotype**	ε2, ε3	NA	2 (16.7)	5 (15.2)	0.9368
	ε3, ε3	NA	7 (58.3)	18 (54.5)	
	ε3, ε4	NA	3 (25.0)	9 (27.3)	
	ε4, ε4	NA	0 (0)	1 (3.0)	
**Comorbidity ***		NA	2 (16.7)	15 (45.5)	0.0962
**Vaccination status**	Vn	NA	1 (8.3)	2 (6.1)	0.7995
	Vb	NA	2 (16.7)	9 (27.3)	
	Va	NA	9 (75.0)	21 (63.6)	
	Vu	NA	0 (0)	1 (3.0)	
**Days till visit, Mean (SD)**		NA	297 (189)	311 (192)	0.5990
**Age in years, Mean (SD)**		47 (14)	39 (14)	45 (13)	0.3060

Data shown are number (%) of patients for all categories except for days till visit and age, which are group means. Abbreviations: NA, not available or not applicable; Vn, unvaccinated; Vb, infected before vaccination; Va, infected after vaccination; Vu, vaccinated but vaccination date is unknown. * Comorbidities are hypertension, obesity, diabetes, lung disease or clear cell carcinoma. Sex, race, ethnicity, APOE genotype and vaccination status were tested using chi-square tests. Comorbidity presence was tested using Fisher’s exact test. Group medians for days till visit were compared using Mann–Whitney U test. Group medians for age were compared using the Kruskal–Wallis test followed by post-hoc Dunn’s test.

## Data Availability

The raw data supporting the conclusions of this article will be made available by the authors on request.

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
