# Peer review of "Blood Markers Show Neural Consequences of LongCOVID-19"

_cells, 2024, doi:10.3390/cells13060478_

Round 1

Reviewer 1 Report

Comments and Suggestions for Authors

Dear Authors,

Please find in the attached document my comments and suggestions.

Best regards,

Reviewer 2 Report

Comments and Suggestions for Authors

In their endeavor to identify neurological biomarkers of LongCOVID-19, Tang et al. investigated the soluble proteins present in the plasma as well as the proteins associated with plasma neuronal enriched extracellular vesicles (nEVs) in 33 LongCOVID-19 patients with neurological impairment (nLongC), 12 COVID-19 survivors without any LongCOVID-19 symptoms (Cov), and samples from 28 pre-COVID-19 healthy controls (HC). The authors reported alterations in IL-1beta, IL-8, BDNF and cortisol in nLongC and Cov compared to HC, indicative of chronic peripheral inflammation with increased stress following COVID-19 infection. Interestingly, toxic proteins associated with neurodegeneration, including amyloid beta 42, pTau181 and TDP-43 were increased in nEVs from people with nLongC. Overall, the study is well-written, some minor suggestions are as follows:

1.        The authors may consider re-writing the first sentence (Lines 35-37) of ‘Introduction’, it may be misconstrued/misunderstood as written. WHO declared the end of COVID-19 because the number of new cases were on decline, not the other way around.  

2.        The authors have recognized the limitations of the study in terms of small sample size and lack of a more standardized cognitive health assessment of people involved in the study. A larger cohort and longer follow-up are needed to conclusively determine the observations of this study.

3.        The observations made in this study are of importance and lay the foundation for further investigations.

4.        Some typographical errors exist warranting a careful reading of the manuscript by the authors.

Comments on the Quality of English Language

English language is fine, some typos exist.

Round 2

Reviewer 1 Report

Comments and Suggestions for Authors

Dear Authors,

Many thanks for implementing my previous suggestions.

I noticed that there are still 15 mentions of the term "biomarker" (in text, title, and abstract). May I kindly ask you to examine the manuscript once more, reassessing the necessity of the use of "biomarker" and, where possible/appropriate, change the term to the word "protein" as you suggested in your reply.

Subsequently, I'm happy to support the publication of your research article in Cells and congratulate you for your work.

Thank you & best regards,

Author Response

3/1/24 Revisions to Manuscript (highlighted in teal color)

Because corrected data for the tTau graph in figure 5 now shows tTau is also upregulated in the nLongC group (in addition to the Cov group), the following minor changes in the text are needed:

Line 19:  Changed word “biomarkers” to “markers” (per Reviewer 1 comment on the use of the term biomarker).

Line 79-80:  Changed from “…while two were significantly…” to “while one was significantly…”

Line 80:  The word “tTau” was deleted since this protein is on longer unique to Cov. 

Line 252:  Changed words “Protein Biomarkers” to “Cytokines” (per Reviewer 1 comment on the use of the term biomarker).

Line 317:  Deleted the word “biomarker” from the phrase “…protein biomarker analyses.” (per Reviewer 1 comment on the use of the term biomarker).

Line 327:  Changed the word “Biomarkers” to “Markers” (per Reviewer 1 comment on the use of the term biomarker).

Line 333:  Deleted the words “and tTau” from the phrase “…, except Ab40 and tTau, …” such that the phrase now reads “…, except Ab40, …”.

Lines 335:  Changed the sentence from “In nEVs from the Cov group, Ab40 and tTau were significantly elevated compared to HC.” to now read “In nEVs from the Cov group, Ab40 was significantly elevated compared to HC.”

Line 336:  Deleted the words “and Tau” from the phrase “Ab40 and tTau” so that new sentence reads “nEVs from people with nLongC had Ab40 levels similar to HC”.

Line 336:  Added the word “tTau” to the sentence that begins with “NRGN and NCAM-1 were …” such that sentence now reads “tTau, NRGN and NCAM-1 were …”.

Line 339:  Added the word “tTau” to the sentence that reads “…, TDP-43 and HMGB1.” Such that the sentence now reads “…, TDP-43, tTau and HMGB1.”

Figure 5.  Changed tTau graph to reflect corrected data values.

Line 348:  Changed word “Biomarker” to “Blood Markers” (per Reviewer 1 comment on the use of the term biomarker).

Line 350:  Changed word “biomarkers” to “markers” (per Reviewer 1 comment on the use of the term biomarker).

Line 352:  Changed from “9 differentially expressed proteins proteins shared by both Cov and nLongC…” to “10 differentially expressed proteins proteins shared by both Cov and nLongC…”.

Line 354:  Added “tTau” to list of shared proteins.

Line 354:  Changed phrase from “… 2 unique to Cov (Ab40 and tTau) …” to “1 unique to Cov (Ab40)”.

Line 356:  Changed word “biomarkers” to “Blood Markers” (per Reviewer 1 comment on the use of the term biomarker).

Line 358:  The sentence was changed from “…with the exception of the bottom 2 proteins, Ab40 and tTau, which were most intense …” to “…with the exception of Ab40 (bottom row), which was most intense …”.

Lines 360, 364, 368:  Changed word “biomarkers” to “markers” (per Reviewer 1 comment on the use of the term biomarker).

Figure 6A.  tTau was moved from the left side to the middle/intersection to reflect its upregulation is no longer unique to Cov but also observed in nLongC.

Figure 6B.  The row for tTau was eliminated as it is no longer unique to either Cov or nLongC.

Line 415:  Changed word “biomarker” to “protein” (per Reviewer 1 comment on the use of the term biomarker).

These are the locations (highlighted in green) where the term “biomarker” has been kept because this is the term used in the original citation: lines 62, 66, 417, 465 (used biomarker term in our original Cells paper).

Line 515:  The word “biomarkers” was kept here because this sentence is defining this term.

The term “biomarkers” was used in seven of the references and, hence, were not changed.